# Sucrose Stearates Stabilized Oil-in-Water Emulsions: Gastrointestinal Fate, Cell Cytotoxicity and Proinflammatory Effects after Simulated Gastrointestinal Digestion

**DOI:** 10.3390/foods13010175

**Published:** 2024-01-04

**Authors:** Danhong Zheng, Weiyan Guan, Jiaqing Chen, Cuicui Zeng, Shen Tan, Jing Chen, Da Ma

**Affiliations:** 1Institute for Advanced and Applied Chemical Synthesis, Jinan University, Guangzhou 510632, China; 2College of Pharmacy, Jinan University, Guangzhou 510632, China; 3College of Packaging Engineering, Jinan University, Zhuhai 519070, China

**Keywords:** sucrose stearate, digestion behavior, lipolysis, bioactivity, GI tract

## Abstract

Different structural composition ratios of sucrose stearates with hydrophilic–hydrophobic balance (HLB) values ranging from 1 to 16 on lipolysis in emulsion were investigated using a simulated gastrointestinal tract (GIT). Results showed a direct correlation between the HLB values of sucrose stearates and the lipolysis rate of emulsions, and a lower HLB value led to diminished lipolysis in the GIT simulation model. Mechanism study indicated that poor emulsifying capacity of sucrose stearates and lipolysis of sucrose stearates with lower HLB value inhibited the digestive behavior of oil. In addition, monoester was mainly hydrolyzed in the gastric phase, whereas sucrose polyesters caused lipolysis in the intestinal phase using an in vitro digestive model and HPLC analysis, further suppressing lipid digestion. Furthermore, a decrease in cell cytotoxicity and proinflammatory effects on Caco-2 and Raw264.7 were observed post-digestion, respectively. This work offers important insights into the effects of the degree of esterification of sucrose stearate on lipid digestion behavior in oil-in-water emulsions.

## 1. Introduction

The digestion behavior of lipids in oil-in-water (O/W) emulsions is highly affected by the extent of lipase adsorption at the oil–water interface [1,2]. The composition and properties of emulsions directly influence the release rate of free fatty acids (FFA). Such factors include the stability of emulsions in the gastrointestinal tract (GIT), the interfacial displacement of emulsifiers by bile salts and lipases, the potential hydrolysis of exogenous emulsifiers at the interface, and the micellization and transport of lipolysis products [1,3]. In the GIT process, partially digested O/W emulsions enter the small intestine and mix with bile and pancreatic juice, influencing the composition, structure, and properties of the interfacial layer [4]. Studies on emulsified lipid digestion suggest that smaller particle sizes stabilized by O/W emulsion emulsifiers, along with various surfactant elements like high bile salts, phospholipids, and proteins, enhance lipase digestion [5]. Thus, exogenous emulsifiers’ physicochemical properties and behavior are largely related to the lipid digestion rate of emulsions in the GIT. Sucrose esters are special non-ionic surfactants made by mixing sugars and fatty acids using chemicals or enzymes. Their hydroxyl groups, strategically placed at eight spots, give them unmatched flexibility to mix with fatty acids, making it easy to adjust the length of the fatty acid chain. This flexibility allows the creation of various sucrose esters with hydrophilic–lipophilic balance (HLB) values ranging from 1 to 18 [6]. Importantly, these sucrose esters are key in oil-in-water emulsions. Their ability to stabilize the crystallization of both oil and water greatly improves the freeze–thaw stability of water-in-oil emulsions [7]. This crucial role makes sucrose esters vital in many areas, especially in food, cosmetics, and pharmaceuticals, where they are essential as emulsifiers and stabilizers [8,9]. Recent studies have indicated that the addition of sucrose esters influences the digestion of lipid emulsions in the GIT, and lipolysis is significantly decreased in emulsions stabilized by sucrose esters in simulated GIT models [10,11]. However, little is known about the lipid digestion behavior and associated mechanism of sucrose ester stabilized emulsions at the GIT.

The HLB values of sucrose stearates provide a prediction of their water solubility and are mainly determined by the degree of esterification (mono-ester content). The structural features of sugar esters, in turn, influence their physicochemical properties, stability, and biodegradability under GIT conditions. Previous studies have shown that mono-esters can be hydrolyzed into sugar and fatty acids directly, and this process is affected by temperature and water in vitro [12]. The stability and emulsifying properties of mono-esters decreased as the length of the carbon chain of the fatty acid component decreased under acidic conditions [13,14]. When there was a sulfonyl or alkyl group adjacent to the ester moiety of the sugar ester, they did not undergo hydrolysis but instead were degraded by initial alkyl chain oxidation [15]. Furthermore, the degree of esterification of sucrose esters also directly influences their ability to undergo hydrolysis; a higher degree of esterification creates steric hindrance within the sugar structure, thus inhibiting the hydrolysis [12,16]. Therefore, a systematic investigation of the influence of the degree of esterification of sucrose esters on their hydrolysis in vitro and in vivo is likely to provide a clearer understanding of the lipid digestion behavior of the resulting emulsion at the GIT.

Sucrose esters are known to possess a wide range of biological activities, including anti-bacterial [17], cytotoxic [18], anti-fungal [19], and α-glucosidase inhibition [20]. The bioactivity of sucrose esters is affected by their digestion properties in vivo. Monoesters or polyesters cannot be absorbed directly through intestinal mucosa in vivo. Evidence suggests that the biodegradation of sucrose esters into sucrose and fatty acids leads to the alteration of bioactivity and utilization under GIT [21]. Thus, a study on the digestive behavior of sucrose esters will lead to a better understanding of their bioactivity and safety in vivo. In this study, commercial sucrose stearates (Table A1) of varying monoester and polyester content with HLB values ranging from 1 to 16 were selected as emulsifiers to stabilize O/W emulsions to (ⅰ) assess the effect of different sucrose esters on lipid digestion under GIT conditions, and (ⅱ) investigate the digestion behavior and cytotoxicity to elucidate the lipolysis mechanism of the emulsions and evaluate any alterations to their bioactivity under these conditions.

## 2. Materials and Methods

### 2.1. Reagents and Chemicals

Sucrose stearates S-170, S-270, S-570, S-770, S-970, S-1170, S-1570, and S-1670 were purchased from Mitsubishi-Chemical Foods Corporation (Tokyo, Japan); 6-*O*-stearoylsucrose (SSE, >98%) was synthesized enzymatically in our laboratory using the standard method. Pancreatic lipase, mucin, bile salt, pepsin, and 3-(4,5-dimethylthiazol-2-yl)-2,5-diphenyltetrazolium bromide (MTT) were purchased from Sigma (St Louis, MO, USA). n-Hexadecane and Tween-80 were purchased from Aladdin (Shanghai, China). The Caco-2 and Raw264.7 cell lines were purchased from Procell Life Science & Technology Co., Ltd. (Wuhan, China) and cultured in DMEM with 10% fetal bovine serum and 1% penicillin/streptomycin (P/S) (Life Technologies, Grand Island, NY, USA) at 37 °C with 5% CO_2_.

### 2.2. Emulsion Preparation

The preparation of emulsion was followed by Andrade, Jonathan et al. with slight modification [22]. Briefly, 0.2 g of S-170, S-270, S-570, S-770, S-970, S-1170, S-1570, S-1670, SSE, and Tween-80 were dissolved in 2 mL of corn oil or n-hexadecane. The corn oil and n-hexadecane solutions were then dispersed in a 98 mL aqueous solution using a high-speed disperser for 2 min (IKA T18 digital, IKA Works Guangzhou Co., Ltd., Guangzhou, China, 1.3 × 10^4^ rpm/min). Subsequently, the dispersed solution was emulsified with a high-pressure homogenizer (APV-1000, Albertslund, Denmark, 900 Psi for 3 cycles).

### 2.3. Droplet Size and Charge Measurements

The mean droplet diameter and ζ-potential of emulsions stabilized by S-170, S-270, S-570, S-770, S-970, S-1170, S-1570, S-1670, and SSE or digestive fluids were measured by dynamic light scattering (DLS) using a Zetasizer Nano-ZS apparatus (Malvern Instruments, Worcestershire, UK). Measurements were performed in triplicate at 25 °C.

### 2.4. Study on the Lipid Digestive Profiles of Emulsions Stabilized by Sucrose Stearate In Vitro

The digestion behavior of corn oil in emulsions stabilized by S-170, S-270, S-570, S-770, S-970, S-1170, S-1570, S-1670, SSE, and Tween-80 were examined using a stimulated gastrointestinal digestion model as described previously with minor modifications [23]. Firstly, a 20 mL emulsion aliquot was mixed with an equal volume of prepared artificial saliva working solution containing 15 mg/mL porcine gastric mucin (type II) and incubated at 37 °C for 2 min. Then, 25 mL of oral reaction fluid was added into an equal volume of simulated gastric fluid working solution, which is made of 1.6 mg/mL pepsin in PBS. The mixture (pH 2.5) was shaken and reacted at 37 °C for 2 h. Then, the pH of 30 mL of gastric reaction fluid was adjusted to 7.0, and 1.5 mL of simulated intestine fluid salt, 3.5 mL of bile salt solution, and 2.5 mL of a solution of 1.6 mg/mL lipase were subsequently added to the sample. The pH of the system was monitored and controlled using an automatic titration unit (pH-stat, Metrohm USA, Inc., Westbury, NY, USA). The reaction system was maintained at 7.0 by titrating a 100 mM NaOH solution into the mixture throughout the 2 h incubation period of the small intestine digestion phase at 37 °C. The percentage of free fatty acid (FFA) released was calculated based on the resulting titration curve using the following equation [24]:$$FFA\left(\%\right)=\frac{V_{NaOH}m_{NaOH}M_{lipid}}{2\;W_{lipid}}\times 100$$
where V_NaOH_ and m_NaOH_ are the volume and molarity of the NaOH solution added, respectively, and M_lipid_ and W_lipid_ are the molecular weight and weight of the lipids (corn oil) in the small intestine phase, respectively.

### 2.5. Confocal Microscopy

The emulsions stabilized by S-170, S-570, and S-1670 and their gastric fluids, as well as intestinal fluids (500 µL), were mixed with 10 μL of Nile Red stock solution (100 *w*/*v*% dissolved in ethanol), respectively. A small quantity of the solution was placed on a 3.5 cm glass-bottom dish, and the microstructure of the stained solution was observed using a confocal laser scanning microscope (ZEISS LSM 880, Carl Zeiss Inc., Oberkochen, Germany).

### 2.6. Study on the Digestive Behavior of S-170 to S-1670 in a Simulated GIT

The FFA release percentages of 2% n-hexadecane solution stabilized by 0.2% S-170, S-270, S-570, S-770, S-970, S-1170, S-1570, and S-1670 were determined using the simulated gastric and intestinal working condition as described above. The reaction system was maintained at pH 2.0 in the gastric phase and pH 7.0 in the intestinal phase by titrating with a 10 mM NaOH solution throughout the reaction and was monitored using an automatic titration unit. The amount of FFA released was calculated using the equation shown above.

### 2.7. High-Performance Liquid Chromatography (HPLC) Analysis

The monoester and polyesters of S-170, S-270, S-570, S-770, S-970, S-1170, S-1570, and S-1670 were analyzed and quantified by HPLC after digestion in a simulated GIT. The protocol of HPLC was based on Hubert, Florence, et al. with a slight modification [25]. The freeze-dried samples were fractionated using a Chromolith Fast Gradient reversed-phase column (50–52 mm) in an HPLC system (WAT270944 2695, Waters Corporation, Milford, MA, USA) equipped with an evaporative light-scattering detector (ELSD, Waters Corporation). The column temperature was held at 30 °C. The ELSD was operated at 55 °C with N_2_ as the nebulizing gas at a pressure of 40 psi. The injection volume was 10 µL, and the solvent flow rate was set at 0.3 mL/min. Gradient elution was carried out as follows: 0 min–15 min, 75% methanol, 25% water; 15 min–60 min, 100% methanol. The calibration curves of monoester stearate and sucrose were prepared using the purified monoester stearate and sucrose.

### 2.8. Cell Viability

The MTT assay was carried out as described previously with minor modifications [26]. Briefly, Caco-2 cells were seeded onto a 96-well plate (0.3 × 10^4^ cells /well) and cultured overnight. Different concentrations of S-570, S-770, S-970, S-1170, S-1570, S-1670, and SSE were added to the Caco-2 cells and incubated for 72 h. The total amount of 20 μL of MTT (5 mg/mL) was added into each well and incubated for an additional 4 h. Then the supernatant medium was removed, and dimethyl sulfoxide (DMSO, 200 μL) was added to dissolve the formazan crystals formed. The absorbance was determined spectrophotometrically at 570 nm using a microplate reader (BioTek, Winooski, VT, USA). The percentage of cell viability was determined by comparing the absorbance of the control sample with DMSO treatment.

### 2.9. Nitric Oxide (NO) Assay

Raw264.7 cells were seeded into a 96-well plate (4 × 10^4^ cells/well) and cultured overnight. Then, the cells were treated with different concentrations of S-570, S-770, S-970, S-1170, S-1570, S-1670, and SSE for 24 h. LPS (100 ng/mL) was used as the positive control (100%). The amount of NO in each sample was determined using an NO assay kit (Promega, Madison, WI, USA) following the manufacturer’s protocol. Thus, 50 μL of supernatant medium from the different samples was collected. Then, enzymatic conversion of the supernatant nitrate to nitrite by nitrate reductase was determined at 570 nm using a microplate reader (BioTek, Winooski, VT, USA). The percentage of NO released in each sample was calculated as follows:$$NO\;release(\%)=\frac{{OD}_{sample}-{OD}_{blank}}{{OD}_{LPS}-{OD}_{blank}}\times 100$$

### 2.10. Statistical Analysis

The data are presented as mean ± standard deviation (SD). Mean differences were analyzed using one-way analysis of variance (ANOVA) (SPSS v.24 software, IBM, New York, NY, USA), and *p*-values of less than 0.05 were accepted as statistically significant.

## 3. Results and Discussion

### 3.1. The Effect of Sucrose Stearate with Different HLB Values on Lipid Digestion in Emulsion under GIT

The sucrose stearates S-170, S-270, S-570, S-770, S-970, S-1170, S-1570, and S-1670 are comprised of 1% to 75% of the monoester and polyester compounds with hydrophile–lipophile balance (HLB) values ranging from 1 to 16, respectively (Table A1). The particle size and ζ-potential of 2% corn oil emulsions stabilized with S-170, S-170, S-570, S-770, S-970, S-1170, S-1570, and S-1670 were determined as described above (Figure 1A,B). The diameter of the initial emulsions decreased from 0.674 ± 0.023 µm to 0.374 ± 0.034 µm, and the ζ-potential decreased from −18.86 ± 1.33 mV to −66.93 ± 2.37 mV when stabilized by S-170 to S-1670, respectively. A smaller droplet size and lower ζ-potential value indicate an increase in the stability of the emulsion due to the electrostatic repulsion between oil droplets that prevents the emulsion from flocculation [27]. The droplet size data of emulsions suggest that the HLB value of sucrose stearates might play a key role in the colloidal stability of emulsions. As reported in a previous study, higher HLB values corresponded to better stability of oil droplets in the aqueous phase [28]. Emulsifiers with higher HLB values possess more hydrophilic groups. Hence, sucrose stearates with higher HLB values exhibited better emulsifying and dispersive capacity in O/W emulsion. Yinan Zhao et al. have reported that liposome-mediated gene delivery stabilized with higher HLB values of sucrose esters exhibited particle size and ζ-potential, which showed a similar trend to this study [29]. Hence, the lipolysis rate promoted a more stable emulsion, which was stabilized by an emulsifier with a higher HLB value. The diameter of different emulsions significantly increased in the stomach stage. In particular, the droplet size of the S-570, S-270, and S-170 emulsions dramatically increased from 0.609 ± 0.005 µm to 3.745 ± 0.043 µm, 0.623 ± 0.014 µm to 4.905 ± 0.051 µm, and 0.674 ± 0.023 µm to 5.084 ± 0.241 µm, respectively, after gastric digestion. The type and stability of emulsifiers contribute to the physiochemistry characteristics of emulsions under stimulated GIT. A significant destabilization of oil droplets has been observed in emulsions stabilized by whey protein isolates or caseinates under gastric conditions [30]. In this study, we observed that the stability of the emulsions decreased with decreasing HLB values of the sucrose stearates in the gastric phase. The net charge of the gastric fluids reduced significantly to approximately zero as the pH value approached 2.5. Previous studies have shown that the degree of ionization of sucrose esters at the oil–water interface is lowered in acidic environments [31]. In the intestinal phase of the digestive process, the diameter and negative charge of the fluid samples increased further without obvious regular change as a result of the neutral pH environment and the presence of bile salts in the intestinal phase.

The lipid digestion profile of the samples in the intestinal phase is shown in Figure 1C. Our results show that the FFA release rate decreased as the HLB value of the sucrose stearates decreased. In S-170, S-270, and S-570 (99%, 90%, and 70% polyester content, respectively) stabilized emulsions, the degree of digestion of corn oil reduced significantly compared with emulsions stabilized by S-770, S-970, S-1170, S-1570, and S-1670. Thus, a higher ratio of polyester to monoester within the sucrose stearate samples suppressed lipolysis in O/W emulsions under gastrointestinal conditions. The confocal microscopy results showed that the lipolysis rate of emulsions stabilized by these samples decreased in the order S-1670 > S-570 > S-170, both in the gastric and intestinal phases (Figure 2). The red fluorescent section indicated the lipids in the emulsion droplets. For the initial stage, the droplet lipid was small and well dispersed, stabilized by S-1670, S-570, and S170, respectively. Studies have shown that lipid digestion begins in the stomach with the action of gastric lipase and occurs primarily in the intestine [2,32]. The red fluorescent images (Figure 2) indicated that while a small portion of lipid was digested in the stomach, it was primarily digested in the intestine. In addition, lipolysis behavior in vivo is influenced by the type of emulsifiers on the emulsion interface [33,34]. Smaller droplet lipids were observed in the emulsion stabilized by S-1670 at the gastric digestive stage. Emulsions stabilized by sucrose esters with higher HLB values have a smaller droplet size, resulting in a higher rate of displacement of the emulsifiers from the droplet surface by bile salts. In these emulsions, the increased interfacial contact between the oil droplet and lipase contributed to a higher efficiency of lipid digestion in the intestine [31]. Thus, the monoester/polyester ratio of the sucrose stearate in stabilized emulsions directly affects FFA release in a simulated GIT model, and at higher degrees of esterification of the sucrose ester, FFA release in vitro is suppressed.

### 3.2. The Effect of Different Types of Emulsifiers with the Same HLB Value on Lipolysis Behavior

While studies have shown that emulsifiers with different HLB values have distinct emulsifying abilities that affect lipid digestion., we aim to investigate the effect of different surfactants in emulsifiers with similar HLB values. To this end, Tween-80 (HLB 15), S-1570 (HLB 15), S-1670 (HLB 16), and SSE (HLB 16) were selected as surfactants to study the lipolysis behavior of the resulting emulsions in a simulated GIT model. Our experiments revealed that the particle size of Tween-80 stabilized emulsions remained relatively constant during the gastric and intestinal phases of digestion (Figure 3A), consistent with previous reports [31], whereas the particle sizes of S-1570, S-1670, and SSE stabilized emulsions increased significantly in both gastric and intestinal phases. Studies have shown that emulsion droplet size may vary during digestion depending on the initial droplet size and its interfacial composition [34]. While sucrose esters, like Tween-80, which is a polysorbate, are non-ionic surfactants, unlike Tween 80, they are unstable in acidic environments. This is demonstrated in the observed decrease in the ζ-potential of the sucrose ester (S-1570, S-1670, and SSE) stabilized emulsions during the gastric phase and subsequent increase during the intestinal phase (Figure 3B). Although the HLB values of Tween-80, S-1570, S-1670, and SSE are similar, the ζ-potential of the initial emulsion stabilized by these surfactants, viz −45.47 ± 1.63 mV, −52.03 ± 1.07 mV, −56.96 ± 2.23 mV, and −49.70 ± 1.3 mV, respectively, were considerably different. While the negative charge of the emulsions stabilized by S-1570, S-1670, and SSE decreased significantly in gastric fluids (pH 2.5), these figures increased in the intestinal phase (pH 7.0). In addition, the FFA release of all four emulsions was investigated and was found to quickly increase for the first 10 min during the intestinal phase of digestion (Figure 3C) in the order of Tween-80 > SSE > S-1670 > S-1570. While the HLB values of Tween-80 and S-1570 are the same (HLB 15) and lower than SSE and S-1670 (HLB 16), the lipid digestion rate of the Tween-80 emulsion was the fastest among the samples. Based on these results, we identified that the larger droplet size and a slower rate of lipolysis observed for the emulsions stabilized by the sucrose esters were a result of the hydrolysis of the sucrose stearate in the gastric phase. In addition, the high HLB values of the sucrose esters led to a higher tendency for these emulsifiers to be displaced by bile salts, decreasing contact between the interface of the droplet and lipase, thus inhibiting FFA release [35]. Hence, our results show that the type of emulsifier, and not just their HLB values, is important when considering the digestion behavior of stabilized emulsions; the hydrolysis of sucrose esters at the interface of the emulsion droplets plays an important role in lipid digestion. Verkempinck et al. have previously reported that Tween-80 stabilized emulsions led to a higher extent of lipolysis compared to sucrose ester stabilized emulsions [36]. Although there was no significant difference in the initial oil droplet size, sucrose ester stabilized emulsions led to the formation of coalesced oil droplets after gastric digestion [36]. While the precise mechanism by which sucrose ester hydrolysis affects lipid digestion is yet to be revealed, it can be concluded that the property of the emulsifiers plays an important role in determining the physicochemical property of the emulsion formed and, in turn, influencing its lipid digestion behavior under gastrointestinal conditions.

### 3.3. The Digestive Behavior of a Series of Sucrose Stearate under Gastrointestinal Conditions

Sucrose esters are nonionic surfactants that can be hydrolyzed in low pH environments or by the action of lipases [12,15]. As such, studies concerning the digestion behavior of sucrose esters under gastric and intestinal conditions would aid in the elucidation of their lipid digestion rule in vivo. To this end, n-hexadecane, which cannot be hydrolyzed by lipase, was used to prepare the emulsions [37]. The diameter and ζ-potential of emulsions stabilized by different sucrose stearates at different digestive stages in vitro are shown in Figure 4A,B. The trend of the change in particle size and charge throughout the gastric and intestinal phases observed for the emulsions prepared with n-hexadecane was similar to that of the corn oil preparations; the particle size of the sucrose ester stabilized emulsions increased progressively as they passed through a simulated GIT. The interfacial properties of sucrose ester stabilized emulsions are influenced by the pH of the environment. In the acidic gastric phase, where the pH reaches close to the isoelectric point (pI) of oil droplets, the instability of the droplets increases [36]. At the same time, protonation of the droplets in the low pH environment of the stomach leads to a decrease in their negative charge, which subsequently increases as they move through the intestine.

The release of FFA in the stomach was determined by measuring the volume of NaOH consumed in the titration process (Figure 4C). Our results indicated that a higher rate of hydrolysis of the sucrose stearate occurred in the gastric phase in samples containing a higher proportion of monoester. After an initial rapid increase in the first 20 min, the release of FFA increased at a slower rate thereafter. At the endpoint (2 h), there was no significant difference between the S-770 and S-570 samples in the amount of FFA released, which were both significantly lower than that of the S-1670, S-1570, S-1170, and S-970 samples and higher than that of S-170 and S-270 samples. This result demonstrated that the sucrose stearates were unstable and hydrolysed in low pH environments, resulting in a lower emulsifying property. As a result, the droplet size of the emulsions stabilized by sucrose stearate increased in the stomach compared to the initial samples (Figure 1A and Figure 3A). Interestingly, during the intestinal phase, FFA release observed in sucrose stearate emulsions increased as the ratio of sucrose polyester increased (Figure 4D). This indicated that the digestion of sucrose polyesters occurred primarily in the intestinal phase by lipase-induced degradation. Given these data, the hydrolysis of sucrose esters affected lipolysis in emulsion due to altering diameter and the interfacial characteristics of the oil droplets in the stomach and intestine (Figure 3 and Figure 4). Furthermore, due to the digestion of sucrose polyesters by lipase), the lipid digestion rate of emulsions stabilized by sucrose stearate with a higher polyester content is reduced (Figure 1C. These results illustrate the mechanism by which sucrose stearates influence lipid digestive behavior; sucrose monoesters are mainly hydrolyzed in the stomach, leading to an increase in droplet size, while sucrose polyesters are digested primarily by lipase in the intestinal phase, resulting in lower digestive efficiency of the oil droplet.

### 3.4. The Quantification of Sucrose Stearates after In Vitro Digestion

The monoester and polyester content of the sucrose stearate is outlined in Table A1. The content of sucrose monoester and polyester in S-170 to S1670 was analyzed after digestion by HPLC. The result is presented in Table 1. The sucrose monoesters S-270 to S-1670 were completely hydrolyzed during passage through a simulated GIT, while 0.33% of S-170 remained after the digestion process. A sucrose peak was observed in all the sucrose ester samples post-digestion, confirming the hydrolysis of the sucrose esters into sucrose and fatty acids. After the intestinal digestion phase, the polyester content remained at 8.19%, 6.39%, 6.45%, 9.75%, 7.40%, 7.25%, 8.26%, and 7.38% in samples S-170 to S-1670, respectively, suggesting that while most of the polyesters were broken down into monoesters and subsequently hydrolyzed into sucrose and fatty acids, a small percentage remained undigested.

Previous studies have shown that ferulic acid sugar esters are hydrolyzed and absorbed completely in the foregut and cecum in rats [38]. In this study, the degradation of sucrose monoesters was found to occur mainly in the stomach, whereas unhydrolyzed sucrose polyesters were digested by lipase in the intestine (Figure 3C,D). The results obtained from HPLC also indicated that >90% of all sucrose esters were digested in a simulated GIT. Due to the hydrolysis of sucrose monoesters, the diameter of gastric digestive emulsion with S-1670, S-1570, and SSE was higher than that of Tween-80 (Figure 3A). In addition, corn oil digestion in emulsion stabilized with sucrose stearate decreased due to the degradation of sucrose polyesters by lipase in the intestinal phase. These results showed that the use of sucrose esters as a surfactant did not just affect the lipid digestion behavior of the emulsions but also altered the bioaccessibility of lipids in vivo. Trends in the digestion behavior of emulsions stabilized by sucrose esters with various HLB values observed in this work may provide some guidance on the utilization of these surfactants in food manufacturing processes.

### 3.5. Bioactivity Studies

The cytotoxicity and inflammatory activity of S-1670 to S-570 before- and post-digestion in vitro were assessed using Caco-2 and Raw264.7 cells (Figure 5A,B). The number of sucrose esters normally added as a surfactant in food production processes was 0.1% to 0.3% (*w*/*v*). Thus, we first assessed cell viability within this concentration range. At 50 mg/mL of S-570 to S-1670, cell viability of Caco-2 was found to range from 61.37 ± 1.75% to 68.03 ± 1.33% before digestion, while it ranged from 88.01 ± 6.17% to 92.97 ± 4.07% after digestion. There was no significant difference among samples at the same dose. The cell viability of Raw264.7 cells ranged from 67.75 ± 4.03% to 74.07 ± 4.24% and 88.20 ± 4.23% to 92.45 ± 8.23% with non-digested or digested sucrose stearates (50 mg/mL) treatment, respectively. These data suggested that the cytotoxicity of sucrose esters was decreased after digestion. Similar observations were seen on the pro-inflammatory effect of the sucrose esters; NO release in Raw264.7 cells decreased after digestion in vitro (Figure 5C). During digestion, sucrose esters are hydrolyzed into sucrose and fatty acids, thus explaining the decrease in their cytotoxicity in vivo. Previous studies have shown that the diesters of S-570 and S-770 exhibited lower cytotoxicity in Caco-2 cells compared to the monoesters [21]. However, S-570 and S-770 were safe for Caco-2 viability after digestion in vitro. The cytotoxicity of SSE is slightly higher than that of S-170 to S-1670 in Caco-2 and Raw264.7 cells, indicating that the cytotoxicity of monoester of sucrose stearate is stronger than that of polyester of sucrose stearate at the same dose, and there was no significant difference between S-570 and S1670. This work confirms the safety of the utilization of sucrose esters as nonionic surfactants in the food industry.

Aside from their widespread use as food additives, including emulsifiers, improvement of starch structure as well as decrease the cholesterol in food, sucrose esters also play an important role in the pharmaceutical industry as enhancers for drug delivery [39]. The degradation of sucrose esters in the GIT affects their bioactivity, including anti-bacterial, anti-fungal, and anti-proliferation of tumor cells when administered orally. Liposoluble nutrient delivery systems that employ sucrose esters as surfactants might be compromised in the GIT, as hydrolysis of the sucrose esters would decrease the release and absorption of liposoluble compounds through the mucosal tissue. Results from this study could provide useful information for future designs of more efficient delivery systems that utilize sucrose ester surfactants.

## 4. Conclusions

This study has shown that the lipid digestion of emulsions is significantly affected by the structural composition of the sucrose ester surfactant. The digestive behavior of oil droplets increased as the polyester of sucrose stearate content increased, in the order S-170, S-270, S-570, S-770, S-970, S-1170, S-1570, and S-1670, under simulated gastrointestinal conditions. The digestive characteristics indicated that the monoesters were hydrolyzed primarily in the gastric phase, while the polyesters were digested by lipase in the intestinal phase. The digestion rate of oil droplets decreased as the ratio of polyester to sucrose stearate increased. Therefore, the lower emulsifying ability and lipolysis behavior of non-ionic sucrose stearate surfactants contribute to the suppression of oil droplet digestion. In addition, the cytotoxicity and proinflammatory effects of S-570, S-770, S-970, S-1170, S-1570, and S-1670 decreased post-digestion. In summary, we have outlined the effect of different sucrose esters with different structural compositions on the lipid digestion behavior of stabilized emulsions in a simulated GIT model.

## Figures and Tables

**Figure 1 foods-13-00175-f001:**
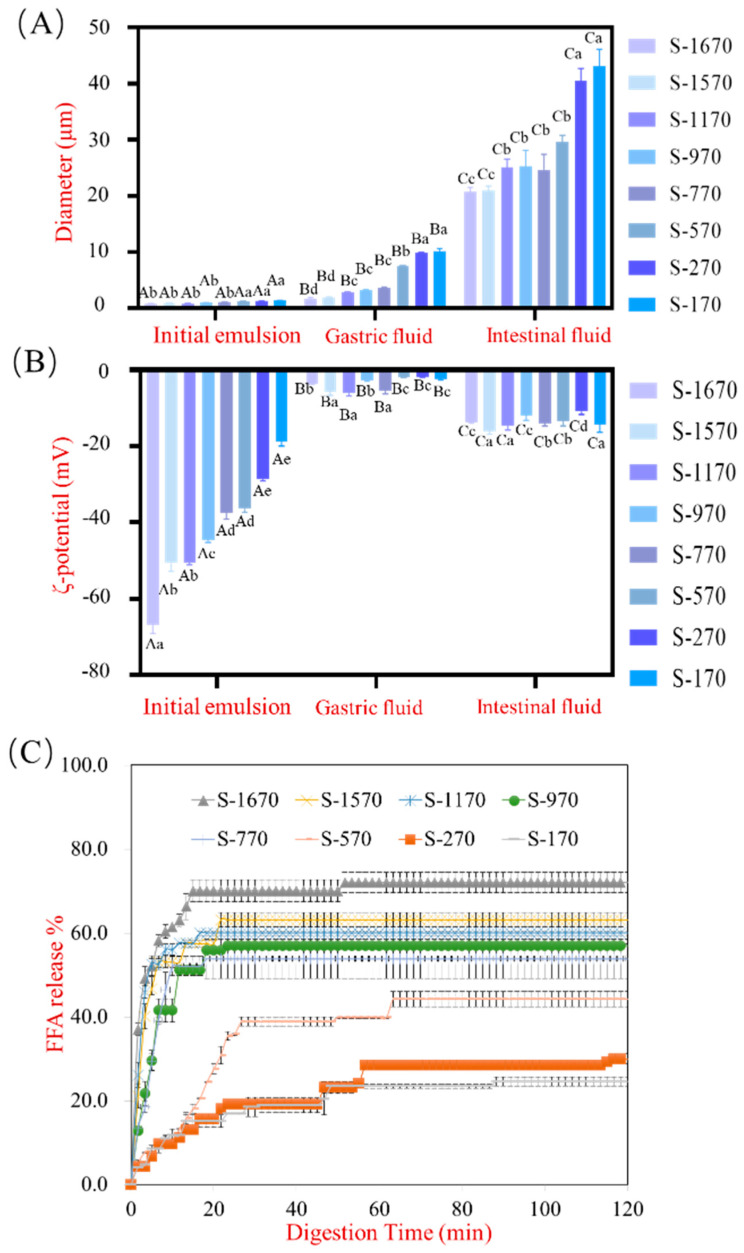
The digestion profile of lipids in emulsions stabilized by sucrose esters under GIT conditions. (**A**) the diameter and (**B**) ζ-potential of emulsions stabilized by S-170, S-270, S-570, S-770, S-970, S-1170, S-1570, and S-1670 were determined in a simulated GIT model. (**C**) release of FFA during digestion from emulsions stabilized by S-170, S-270, S-570, S-770, S-970, S-1170, S-1570, and S-1670 were quantified by the consumption of NaOH volume (100 mM). The values are presented as mean ± SD (*n* = 3). Different lower-case letters indicate statistically significant differences at *p* < 0.05 (one-way analysis of variance followed by Duncan’s test). Superscripted upper-case letters highlight comparisons of the same surfactant intergroup while Superscripted lower-case letters in the same column highlight comparisons of different surfactants intragroup.

**Figure 2 foods-13-00175-f002:**
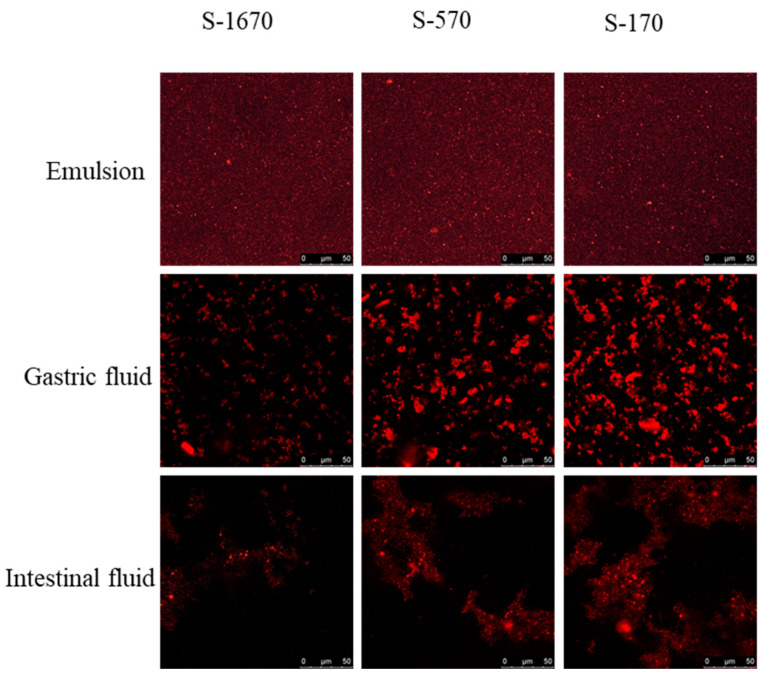
Confocal microscopy images of emulsions after exposure to different GIT stages. The confocal microscopy images of emulsion stabilized by S-1670, S-570, and S-170 were captured after exposure to the gastric and intestinal phases. The pictures were captured at 630× magnification.

**Figure 3 foods-13-00175-f003:**
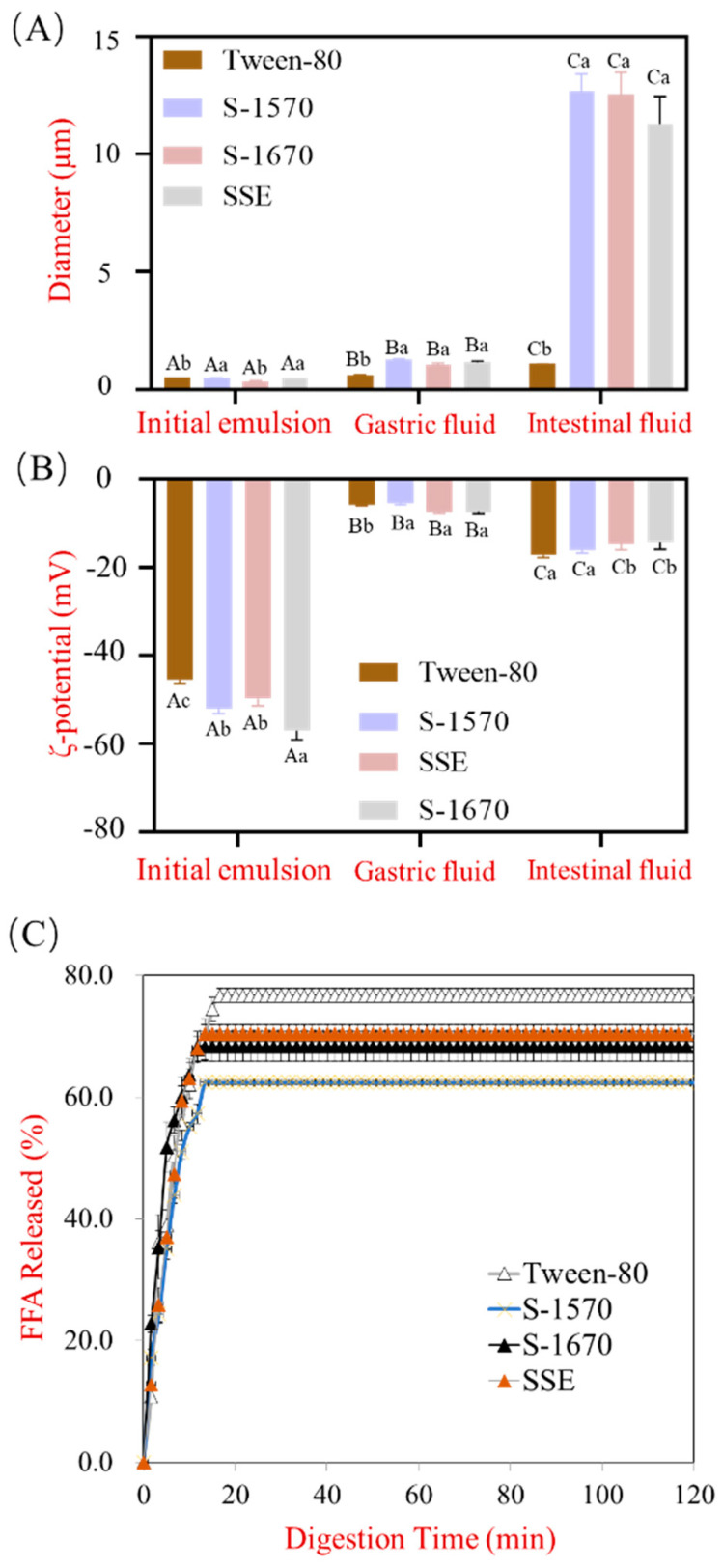
The digestion behavior of emulsions is stabilized by the same HLB of emulsifiers. (**A**) the diameter and (**B**) ζ-potential of emulsions stabilized by Tween-80, S-1670, and SSE were determined after exposure to different GIT stages. (**C**) release of FFA from emulsions stabilized by Tween-80, S-1670, and SSE was determined by the consumption of NaOH volume (10 mM). The values are presented as mean ± SD (*n* = 3). Different lower-case letters indicate statistically significant differences at *p* < 0.05 (one-way analysis of variance followed by Duncan’s test). Superscripted upper-case letters highlight comparisons of the same surfactant intergroup while Superscripted lower-case letters in the same column highlight comparisons of different surfactants intragroup.

**Figure 4 foods-13-00175-f004:**
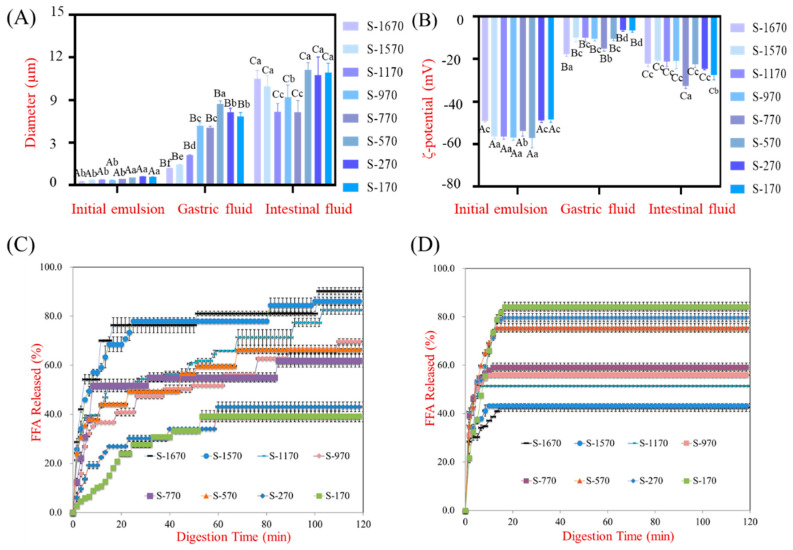
The digestive behavior of S-170, S-270, S-570, S-770, S-970, S-1170, S-1570, and S-1670 after exposure to different digestive stages of a simulated GIT. (**A**) the diameter and (**B**) ζ-potential of 2% n-hexadecane solution stabilized by 0.2% S-170, S-270, S-570, S-770, S-970, S-1170, S-1570 and S-1670 were determined in a simulated GIT model. (**C**,**D**) release of FFA of sucrose stearate esters was quantified by consumption of NaOH volume (10 mM) and NaOH volume (100 mM), respectively. The values are presented as mean ± SD (*n* = 3). Different lower-case letters indicate statistically significant differences at *p* < 0.05 (one-way analysis of variance followed by Duncan’s test). Superscripted upper-case letters highlight comparisons of the same surfactant intergroup while superscripted lower-case letters in the same column highlight comparisons of different surfactants intragroup.

**Figure 5 foods-13-00175-f005:**
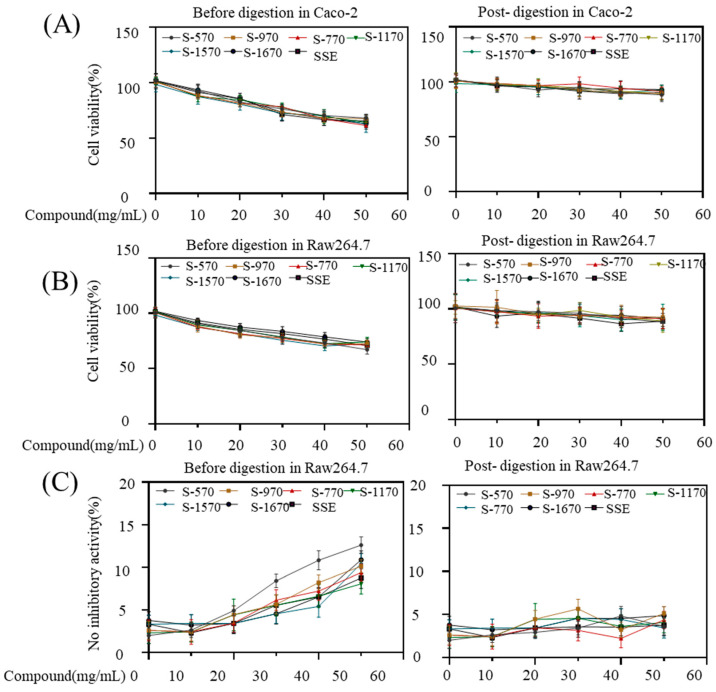
The biological activity of S-570, S-770, S-970, S-1170, S-1570 and S-1670 before or post-digestion. (**A**) Caco-2 and (**B**) Raw264.7 cells (3 × 10^3^ cell/ 96-well plate) were cultured for 24 h and then treated with sucrose esters (0, 10, 20, 30, 40, 50 mg/mL) for 72 h before or after exposure to GIT conditions, respectively. (**C**) Raw264.7 cells (4 × 10^4^ cell/ 96 cm dish) were cultured overnight and then treated with sucrose esters (0, 10, 20, 30, 40, 50 mg/mL) for 24 h before or after digestion. Cell viability and nitric oxide release were determined using a kit described in the Materials and Methods section. The values are presented as mean ± SD (*n* = 3).

**Table 1 foods-13-00175-t001:** Quantification of sucrose stearates before and after digestion by HPLC.

Sample	Initial				Intestinal			
	Monoester (mg/mL)	Monoester (%)	Polyester (mg/mL)	Polyester (%)	Monoester (mg/mL)	Monoester (%)	Polyester (mg/mL)	Polyester (%)
S-170	0.024	1.21%	1.9699	98.49%	0.0066	0.33%	0.1638	8.19%
S-270	0.276	13.82%	1.679	83.95%	0	0	0.1278	6.39%
S-570	0.5144	25.72%	1.478	73.90%	0	0	0.129	6.45%
S-770	0.689	34.45%	1.2956	64.78%	0	0	0.195	9.75%
S-970	0.8634	43.17%	1.1226	56.13%	0	0	0.148	7.40%
S-1170	1.1622	58.11%	0.8322	41.61%	0	0	0.145	7.25%
S-1570	1.3984	69.92%	0.5698	28.49%	0	0	0.1652	8.26%
S-1670	1.558	77.90%	0.393	19.65%	0	0	0.1476	7.38%

## Data Availability

Data are contained within the article.

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
