# Peer review of "Sucrose Stearates Stabilized Oil-in-Water Emulsions: Gastrointestinal Fate, Cell Cytotoxicity and Proinflammatory Effects after Simulated Gastrointestinal Digestion"

_foods, 2024, doi:10.3390/foods13010175_

Round 1
Reviewer 1 Report
Comments and Suggestions for Authors
Reviewer’ Comments
Some important points have to be clarified or fixed before I can proceed and a positive action can be taken.
I hereby summarize these points:
1. It is really unclear to me the meaning of the sentence in the lines 17 and 18 " Different structural composition ratios of sucrose stearates, and hydrophilic-hydrophobic balance (HLB) values on lipolysis in emulsion were investigated " please check this sentence
2. List the methods in the same order as they appeared in the Results section.
3. The description of Figure 2 is not clear. Please add more details.
4. In lines 247, 248, 249, the authors mention that they selected Tween-80 (HLB 15), S-247 1570 (HLB 15), S-1670 (HLB 16), and M (HLB 16) as surfactants to study the lipolysis behavior of the resulting emulsions in a simulated GIT model. However, in the text and in Figure 3, instead of "M (HLB 16)", they talked about "SSE". Please clarify the correct one.
5. Please check the sentence in lines 255-256
6. In Figure 4, there is no description of what is contained (D).

Author Response
Some important points have to beclarified or fixed before I can proceed and a positive action can be taken.
Response: Thanks for the reviewer’s positive comments. The manuscript has been reviewed and improved according to the guidelines provided below.
I hereby summarize these points:
- It is really unclear to me the meaning of the sentence in the lines 17 and 18 " Different structural composition ratios of sucrose stearates, and hydrophilic-hydrophobic balance (HLB) values on lipolysis in emulsion were investigated " please check this sentence
Response: Thanks for your comments. The sentence has been revised that Different structural composition ratios of sucrose stearates with hydrophilic-hydrophobic balance (HLB) values ranged from 1 to 16 on lipolysis in emulsion were investigated. The revised section has been marked in red.
- List the methods in the same order as they appeared in the Results section.
Response: Thanks for your comments. We have changed the method order with red mark. The cell viability section was followed with emulsion digested section.
- The description of Figure 2 is not clear. Please add more details.
Response: Thanks for your comments. More detail information has been added in line 232-234 and line 239-240 with red mark.
- In lines 247, 248, 249, the authors mention that they selected Tween-80 (HLB 15), S-247 1570 (HLB 15), S-1670 (HLB 16), and M (HLB 16) as surfactants to study the lipolysis behavior of the resulting emulsions in a simulated GIT model. However, in the text and in Figure 3, instead of "M (HLB 16)", they talked about "SSE". Please clarify the correct one.
Response: Thanks for your comments. M (HLB 16) have been revised as SSE in line 255.
- Please check the sentence in lines 255-256
Response: Thanks for your comments. This sentence has been revised in line 260 with red mark.
- In Figure 4, there is no description of what is contained (D).
Response: Thanks for your comments. The description of figure 4D was from line322-324.

Reviewer 2 Report
Comments and Suggestions for Authors
The manuscript deals with an interesting subject and aims to elucidate the ‘effects of the degree of esterification of sucrose stearate on lipid digestion behavior in oil-in-water emulsions’. I would suggest the paper need a minor revision. The specific points those should be clarified are listed as below:
· Introduction section does not provide sufficient background about oil-in-water emulsions and sucrose stearates, new references should be added to improve this section.
· The introduction should discuss the relation between lipase adsorption and the composition and properties of emulsions which is connected to the release rate of free fatty acids.
· The authors should further discuss the extent of sucrose ester usage in food industry.
· The Materials and Methods section should be reorganized. Why did you place the cell viability and NO assays prior to emulsion preparation sections?
· The discussion in 3.1 section should be deepen, the authors just presented their results and compared the results with previous studies.
I found the quality of English language acceptable.
Author Response
The manuscript deals with an interesting subject and aims to elucidate the ‘effects of the degree of esterification of sucrose stearate on lipid digestion behavior in oil-in-water emulsions’. I would suggest the paper need a minor revision. The specific points those should be clarified are listed as below:
· Introduction section does not provide sufficient background about oil-in-water emulsions and sucrose stearates, new references should be added to improve this section.
Response: Thanks for your comments. The effect of sucrose stearates on digestion of oil-in-water emulsions have been discussed in line 39-42.
· The introduction should discuss the relation between lipase adsorption and the composition and properties of emulsions which is connected to the release rate of free fatty acids.
Response: Thanks for your comments. The relation between lipase adsorption and the composition and properties of emulsions has been discussed in line 240-244 with red mark.
· The authors should further discuss the extent of sucrose ester usage in food industry.
Response: Thanks for your comments. we have added the extent of sucrose ester usage in food industry in line 409-410.
· The Materials and Methods section should be reorganized. Why did you place the cell viability and NO assays prior to emulsion preparation sections?
Response: Thanks for your comments. We have changed the method order as they appeared in the Results section.
· The discussion in 3.1 section should be deepen, the authors just presented their results and compared the results with previous studies.
Response: Thanks for your comments. we have added the discussed information in 3.1 section in line 193-195 and 198-199.
